# Efficacy and safety of anakinra in adults presenting deteriorating respiratory symptoms from COVID-19: A randomized controlled trial

Alexandra Audemard-Verger[1,2]*, Amélie Le Gouge[3], Vincent Pestre[4], Johan Courjon[5], Vincent Langlois[6], Marc-Olivier Vareil[7], Mathilde Devaux[8], Boris Bienvenu[9], Vincent Leroy[10], Radjiv Goulabchand[11], Léa Colombain[12], Adrien Bigot[1], Thomas Guimard[13], Youcef Douadi[14], Geoffrey Urbanski[15], Jean François Faucher[16], Laurence Maulin[17], Bertrand Lioger[18], Jean-Philippe Talarmin[19], Matthieu Groh[20], Joseph Emmerich[21], Sophie Deriaz[1], Nicole Ferreira-Maldent[1], Ann-Rose Cook[1], Céline Lengellé[22], Hélène Bourgoin[23], Arsène Mekinian[24], Achille Aouba[25], François Maillot[1,2], Agnès Caille[2,3]

1 Department of Internal Medicine and Clinical Immunology, CHRU Tours, Tours, France, 2 University of Tours, Tours, France, 3 INSERM CIC1415, CHRU Tours, Tours, France, 4 Department of Internal Medicine and Infectious Diseases, CH Avignon, Avignon, France, 5 Department of Infectious Diseases, Université Côte d'Azur, CHU Nice, Nice, France, 6 Department of Internal Medicine, CH du Havre, Le Havre, France, 7 Department of Infectious Diseases, CH de Bayonne, Bayonne, France, 8 Department of Internal Medicine, CH de Poissy, Poissy, France, 9 Department of Internal Medicine, Hôpital Saint Joseph, Marseille, France, 10 Department of Infectious Diseases, Clinique Tessier, Valenciennes, France, 11 Internal Medicine Department & Department of Infectious and Tropical Diseases, Nîmes University Hospital, University of Montpellier, Nîmes, France, 12 Department of Infectious Diseases, CH de Perpignan, Perpignan, France, 13 Department of Infectious Diseases, CH la Roche sur Yon, La Roche sur Yon, France, 14 Department of Infectious Diseases, CH Saint Quentin, Saint Quentin, France, 15 Department of Internal Medicine and Clinical Immunology, CHU Angers, Angers France, 16 Department of Infectious Diseases, CHU de Limoges, Limoges, France, 17 Department of Infectious Diseases, CH Aix en Provence, Aix en Provence, France, 18 Department of Internal Medicine, CH de Blois, Blois, France, 19 Department of Internal Medicine and Infectious Diseases, CH de Quimper, Quimper, France, 20 Department of Internal Medicine, Hôpital Foch, Suresnes, France, 21 Department of Vascular Medicine, GH Saint Joseph and Université de Paris, INSERM CRESS 1153, Paris, France, 22 Clinical Research Vigilance Unit, CHRU Tours, Tours, France, 23 Department of Pharmacology, CHRU Tours, Tours, France, 24 Department of Internal Medicine, Hôpital Saint Antoine, Sorbonne Université, Paris, France, 25 Department of Internal Medicine, CHU de Caen, Caen, France

* a.audemard-verger@chu-tours.fr

**Data Availability Statement:** All relevant data are within the paper and its Supporting information files.

## Abstract

### Objective

We aimed to investigate whether anakinra, an interleukin-1receptor inhibitor, could improve outcome in moderate COVID-19 patients.

### Methods

In this controlled, open-label trial, we enrolled adults with COVID-19 requiring oxygen. We randomly assigned patients to receive intravenous anakinra plus optimized standard of care (oSOC) vs. oSOC alone. The primary outcome was treatment success at day 14 defined as

**Funding:** ANACONDA was supported by the foundation of the Tours university hospital through the endowment fund of the university hospital of Tours. Sobi supplied anakinra free of charge. The funders had no role in study design, data collection and analysis, decision to publish, or preparation of the manuscript.

**Competing interests:** The authors have declared that no competing interests exist.

patient alive and not requiring mechanical ventilation or extracorporeal membrane oxygenation.

## Results

Between 27[th] April and 6[th] October 2020, we enrolled 71 patients (240 patients planned to been enrolled): 37 were assigned to the anakinra group and 34 to oSOC group. The study ended prematurely by recommendation of the data and safety monitoring board due to safety concerns. On day 14, the proportion of treatment success was significantly lower in the anakinra group 70% (n = 26) vs. 91% (n = 31) in the oSOC group: risk difference—21 percentage points (95% CI, -39 to -2), odds ratio 0.23 (95% CI, 0.06 to 0.91), p = 0.027. After a 28-day follow-up, 9 patients in the anakinra group and 3 in the oSOC group had died. Overall survival at day 28 was 75% (95% CI, 62% to 91%) in the anakinra group versus 91% (95% CI, 82% to 100%) (p = 0.06) in the oSOC group. Serious adverse events occurred in 19 (51%) patients in the anakinra group and 18 (53%) in the oSOC group (p = 0·89).

## Conclusion

This trial did not show efficacy of anakinra in patients with COVID-19. Furthermore, contrary to our hypothesis, we found that anakinra was inferior to oSOC in patients with moderate COVID-19 pneumonia.

## Introduction

Severe acute respiratory syndrome coronavirus 2 (SARS-CoV-2) emerged in China at the end of 2019 from a zoonotic source [1]. In the course of coronavirus disease (COVID-19), caused by SARS-CoV-2, 10% to 15% of patients present moderate to severe symptoms requiring hospitalization and oxygen support [2]. Three to 5% of these patients develop an acute respiratory distress syndrome (ARDS) requiring admission into an intensive care unit (ICU) in order to receive ventilation support [3]. The RECOVERY trial demonstrated that the administration of dexamethasone (DXM) 6 mg/day for 10 days decreased mortality at day 28 among patients receiving mechanical ventilation or oxygen [4]. However, DXM does not ensure complete efficacy. Indeed, in the RECOVERY trial, mortality at 28 days remained high in both groups (22.9% vs. 25.7% in the DXM and usual care group, respectively). Thus the identification of an effective treatment for COVID-19 remains a major concern and a public health emergency.

The pathogenesis of COVID-19 encompasses a "cytokine storm" which includes pro-inflammatory interleukins (IL-1β, IL-6) and the tumor necrosis factor (TNF-α) [5]. In COVID-19, the deleterious and excessive host immune response plays an important role in the course of the disease and its evolution toward ARDS. Mortality is the consequence of viral driven hyperacute inflammation [6].

Therefore, we hypothesized IL-1β to be a potential therapeutic target for suppressing acute hyperinflammation and therefore treat patients at risk of developing ARDS related to COVID-19. Anakinra is a recombinant, anti-human IL-1 receptor treatment, which can specifically bind to IL-1R and inhibit signal transduction. For up to 20 years, anakinra has mainly been used to treat patients presenting rheumatoid arthritis, gouty arthritis or auto-inflammatory diseases i.e. Still's disease or cryopyrin associated periodic syndrome [7, 8]. Its safety profile is excellent in these diseases. Anakinra is approved for sub cutaneous administration but has been reported to be used with an intraveinous administration and at much higher doses than

used in approved indications e.g, in septic shock and/or haemophagocytic lymphohystiocyto-sis [9, 10].

Previous observational studies have suggested the possible efficacy of anakinra in COVID-19 pneumonia [11–14]. A recent randomized clinical trial (RCT) did not show any efficacy of anakinra, compared to usual care, in patients hospitalized with mild-to-moderate pneumonia [15].

Herein, we aimed to study the efficacy and safety of anakinra, in addition to optimized standard of care (oSOC) compared to oSOC alone, in moderate COVID-19 patients with deteriorating respiratory symptoms presenting an inflammatory component through a multicenter randomized controlled trial (ANACONDA RCT).

## Patients and methods

### Trial design

This study was an open-label multicenter randomized parallel two-group controlled trial. The trial was approved by an ethics committee (CPP Île-de-France VII) and relevant authorities (ANSM). It was prospectively registered (ClinicalTrial.Gov, NCT04364009). Written informed consent was obtained from all the patients or from a legal representative if they were unable to provide consent. The trial protocol is available in S1 File.

### Setting and participants

Patients were enrolled from 20 University and General Hospitals in France. Patients were eligible if they had a confirmed SARS-CoV-2 infection: positive rRT-PCR and/or typical chest or computed tomographic scan of COVID 19 pneumonia and required oxygen therapy defined by either (i) O2 $\geq$ 4L/min to maintain Sp02> 92% and respiratory rate$\geq$ 24/min or (ii) O2$\geq$ 1L/min and oxygen requirement deterioration defined by an increase in oxygen therapy $\geq$ 2L/min to maintain Sp02> 92%. Patients were included if they had an inflammatory component (reactive C-protein $\geq$ 50mg/L) and were treated with antibiotics (according to local practice). Main exclusion criteria were need for mechanical ventilation or O2 > = 11 L/min in order to maintain Sp02> 92%, contra indication to anakinra and concurrent bacterial infection (detailed in S1 File).

### Randomization and blinding

Participants were randomly assigned to receive either anakinra or oSOC alone in a 1:1 ratio through a computer (SAS based) generated randomisation schedule. Randomization was stratified on the baseline C-reactive protein level (<150 vs. $\geq$ 150 mg/L), baseline requirement of oxygen therapy in order to maintain Sp02 > 92% (3–6 liters per min vs. 7–10 liters per min) and corticosteroid therapy at baseline (< vs. $\geq$ 0.5mg/kg/day prednisone) using permuted blocks of random sizes (of 2 to 12) unknown to the investigators. Randomization was performed by trained staff members using a secure, centralized, interactive web-based response system within the 20 days following the onset of COVID-19 symptoms. Due to the emergency nature of the trial and feasibility issues, no placebo for anakinra was manufactured. Study physicians, research staff, participants, and data analysts were aware of treatment allocation.

### Interventions

The experimental treatment was anakinra 100mg/0.67 mL solution for injection in pre-filled syringes (Sobi, SE-112 76 Stockholm—Sweden). Before administration, the full contents of the prefilled, single use syringes (Anakinra 100 mg) were diluted in 100 mL of saline solution and

administration was performed immediately after preparation over an infusion period of 60 minutes. The patients received an intravenous injection (IV) of anakinra 400mg/day (100mg IV every 6 hours) during 3 days. Then, the patients received an IV injection of anakinra 200mg/day (100mg every 12 hours) during 7 days. In case of renal failure (eGFR<30ml/min), anakinra was administratd 1 day out 2. The total duration of anakinra therapy was 10 days. Optimized standard of care was provided at the discretion of the sites' clinicians and included treatments authorized by the French Health Ministry, regularly updated as knowledge evolved, including antiviral drugs, hydroxychloroquine, corticosteroids, anticoagulants, hydration, nutrition, extra-renal purification, oxygen therapy and vasopressive drugs.

## Outcome measures and follow-up

Patients were followed up to day 28. Baseline measurements included clinical, radiological and biological measurements. Patients were followed daily during hospitalization and study visits were performed at Day 3, Day 10, Day 14 and Day 28. The primary endpoint was treatment success at day 14, defined as a patient being alive and not requiring either of the following: invasive mechanical ventilation (IMV) or extracorporeal membrane oxygenation (ECMO). Secondary outcomes included treatment success (same definition as the primary endpoint), clinical status assessed with the WHO Clinical Progression Scale (WHO-CPS) [16], National Early Warning Score (NEW) [17] and biological parameters such as lymphocytes count, C-reactive protein, ferritin, d-dimers and fibrinogen level, all assessed at day 3, day 10, day 14 and day 28. We also assessed overall survival, time to hospital discharge, time to ICU admission, time to ventilatory support, time to oxygen supply withdrawal over the 28-day follow-up. Safety outcomes included adverse events and serious adverse events (SAEs) such as infection, hypersensitivity, hepatic damages and neutropenia.

## Data quality monitoring

Data quality monitoring included both remote data monitoring and on-site monitoring (except for one patient for which monitoring was only performed remotely for technical reasons) implemented by dedicated staff members who were independent of the site investigators, with 100% source data verification performed for all patients recruited at every site for all critical data points.

## Statistical analysis

The final statistical analysis plan is available in S2 File. The treatment success on day 14 was assumed to be 80% in the oSOC group *versus* 93% in the anakinra group. With 80% power, a 5% 2-sided type I error rate and considering the sequential nature of the design with one interim analysis, the maximal required sample size was 240. Patients were analyzed according to their randomization group. For the primary analysis of the primary outcome, missing data were considered as treatment failure. No imputation was made for secondary outcomes. Overall type I error for the primary outcome analysis was to be controlled with the use of the Pocock alpha spending function, but the early termination led to perform all statistical tests with a 2-sided significance level of 0.05. Treatment success on day 14 was reported as proportions in each group and compared using a chi-square test. The difference in proportions between-groups was also estimated with its 95% confidence interval using the Chan-Zhang exact method. A logistic regression was also performed to obtain the relative effect. A sensitivity analysis was performed on patients without missing data (complete cases analysis). An adjustment on stratification variables for randomization was performed in the framework of a

                                                                                 

linear model with identity link function. Pre-specified subgroup analyses were performed using linear models with identity link functions, including interaction terms.

Treatment success proportions at Day 3, Day 10 and Day 28 were estimated in each group and compared using chi-square tests. Differences in proportions were also estimated with their 95% confidence intervals using the Chan-Zhang exact method. Subgroup analyses of the primary outcome included the baseline reactive C-protein level ($<150$ vs. $\geq 150$ mg/L), requirement of oxygen therapy to maintain $Sp02 > 92\%$ (3–6 liters per min vs. 7–10 liters per min), corticosteroid therapy ($<$ vs. $\geq 0.5$mg/kg/day prednisone), d-dimers level ($<$ vs. $\geq 2000$ng/ml) and lymphocytes count ($<$ vs. $\geq 500$/mm$^3$) analysis.

Comparisons of the WHO progression scale at Day 3, 10, 14 and 28 were performed using Cochran-Armitage tests. Overall survival was summarized using Kaplan-Meier curves and compared with a log rank test. The cumulative incidence of patients with ICU admission, ventilatory support and oxygen withdrawal were analyzed using the competing risk approach (Gray test), with death and hospital discharge as competing events. Hospital length of stay was compared between the two groups using the competing risk approach with death as a competing event. Evolution of the NEW score and inflammatory parameters were compared between the two groups using mixed linear models (after log transformation for non-normally distributed variables). Adverse events and serious adverse events were analyzed using chi-square tests.

Data were analyzed with SAS version 9.4 (SAS Institute Inc), and R version 3.3.1 (R Foundation for Statistical Computing).

## Data and safety monitoring board and trial suspension

An independent data and safety monitoring board (DSMB) was set up at the beginning of the study. Its role was to review the results of the planned interim efficacy analysis and review the safety data on a regular basis (at least every 60 patients included). Unscheduled meetings could also be initiated by the DSMB chairperson or trial sponsor in case of urgent safety concerns.

## Role of the funding source

ANACONDA was supported by a private French funding source through the endowment fund of the university hospital of Tours. Sobi supplied anakinra free of charge. Both SOBI and the funders of the study had no role in the study design, data collection, data analysis, data interpretation, writing of the report or the decision to submit the report for publication.

## Results

### Patient characteristics

The trial began enrollment on 27[th] April 2020. On 20[th] May 2020, a first unscheduled DSMB meeting took place, upon the sponsor's request for safety reasons: among the 7 patients randomized in the anakinra group 3 had died in comparison to none among the 7 randomized to the oSOC group. The DSMB recommendation was to continue the trial and meet again if a difference of at least 5 deaths was observed between both groups (to the detriment of the anakinra group). This latter condition was met on 24[th] September 2020 and the DSMB then recommended that recruitment and intervention be stopped for safety concerns. The sponsor decided to discontinue the study on 20[th] October 2020.

When the trial was stopped, 71 patients had been randomly assigned to the anakinra group (n = 37) or oSOC group (n = 34). Among the 37 patients assigned to receive anakinra, one withdrew consent (Fig 1). Demographic and baseline clinical and biological characteristics of patients are described in Table 1. The mean (SD) age of the patients included in the anakinra

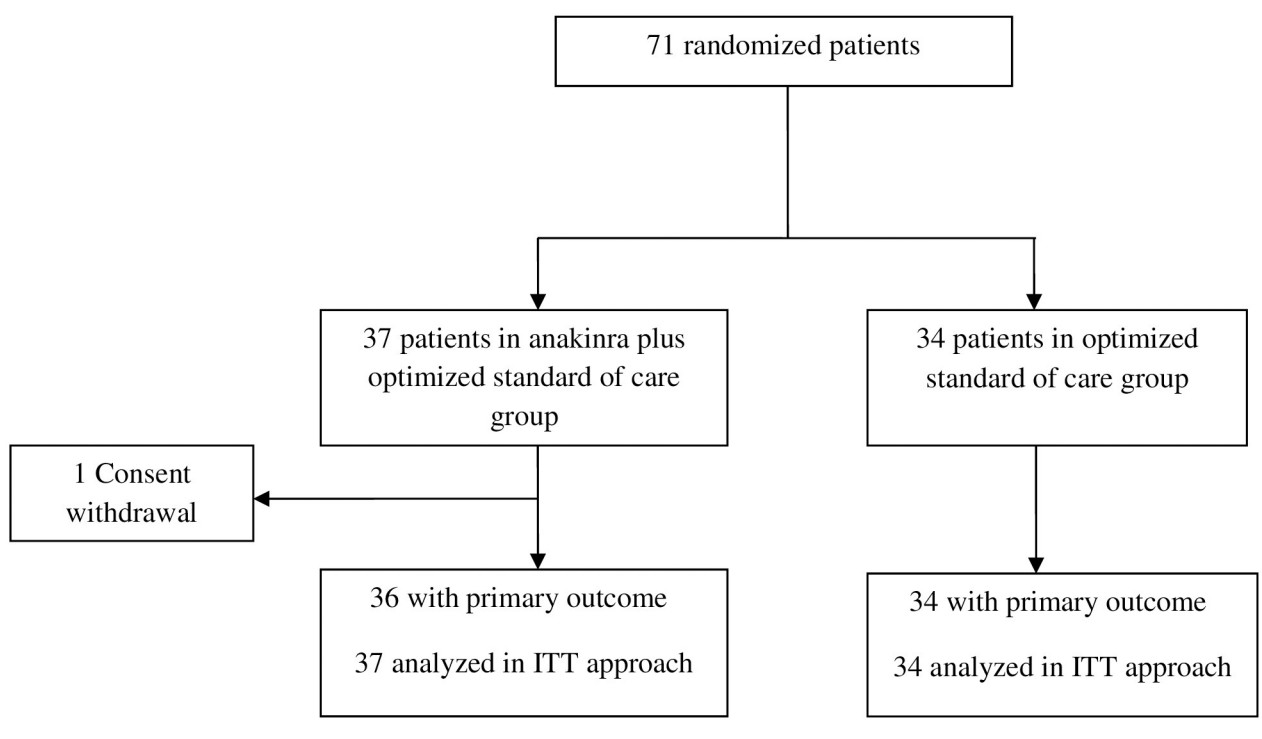

**Fig 1. Flow chart.**

group was 71(14) years, 24% of the patients were female, 97% of the patients had laboratory-confirmed SARS-CoV-2 infection. A history of diabetes was present in 24% of the patients, 54% had high blood pressure. Overall, there was no important between-group differences except for the proportion of patients treated with corticosteroids: 54% in the anakinra group vs. 73% in the oSOC group.

Among the 37 patients assigned to receive anakinra, 1 withdrew consent before the first injection, and 22 did not receive complete anakinra regimen and missed at least one dose of treatment because of death (n = 3), adaptation to renal clearance (n = 2), technical problems (n = 4), adverse events (n = 3), hospital discharge (n = 2), unknown reason (n = 3), decision of the sponsor (n = 2), decision of the clinicians (n = 2) or patient transfer (n = 1) (detail in S1 Table). During the trial, 28 (76%) of patients in the anakinra group and 30 (88%) in the oSOC group received corticosteroids administrated at/or after inclusion. The distribution of mean corticosteroids daily dose did not differ between the 2 groups: 50mg [40; 80] equivalent prednisone/d in the anakinra group vs. 46mg/d [40; 75] in the oSOC group (S2 Table).

## Primary outcome

On day 14, the proportion of treatment success was significantly lower in the anakinra group 70% (n = 26) vs. 91% (n = 31) in the oSOC group; risk difference -21 percentage points; (95% CI, -39 to -2), odds ratio 0.23 (95% CI, 0.06 to 0.91), p = 0.027 (Table 2). The results were similar in the sensitivity analysis (72% (n = 26/36) in the anakinra group vs. 91% (n = 31/34) in the oSOC group alone (risk difference -19 percentage points; [95% CI, -38 to -0.1], p = 0.04) and after adjustment on stratification variables (risk difference percentage points -19; [95% CI -38 to -0.2], p = 0.047). Regarding subgroup analysis (C-reactive protein, oxygen level, corticosteroids therapy, d-dimers level and lymphocytes count), the effect of the intervention was consistent across prespecified subgroups (S1 Fig).

**Table 1. Baseline characteristics of participants.**

| | Anakinra plus oSOC group (n = 37) | oSOC group (n = 34) |
|---|---|---|
| Female, *n* (%) | 9 (24%) | 10 (29%) |
| Age, *y* | 71 (15) | 70 (14) |
| Respiratory rate, *breaths per min* | 25 [20; 28] | 24 [20; 27] |
| SpO$_2$, % | 95 [93; 96] | 94 [92; 95] |
| Oxygen flow, L/min* | 5 [4; 6] | 4 [4; 6] |
| BMI, *kg/m$^2$* | 28 [24; 33] | 28 [25; 32] |
| Time from symptoms onset, *days* | 9 [7; 11] | 9 [7; 11] |
| **Diagnosis of SARS-CoV2 infection** | | |
| Positive RT-PCT | 36 (97%) | 31 (91%) |
| Typical chest CT | 33 (89%) | 32 (94%) |
| **Coexisting conditions** | | |
| Current or former smoker | 15 (44%) | 16 (48%) |
| Diabetes | 9 (24%) | 6 (18%) |
| High blood pressure | 20 (54%) | 15 (44%) |
| Cirrhosis | 0 (0%) | 0 (0%) |
| Coronary disease | 2 (5%) | 7 (21%) |
| Chronic obstructive pulmonary disease | 1 (3%) | 6 (18%) |
| Asthma | 2 (5%) | 1 (3%) |
| Cancer | 1 (3%) | 1 (3%) |
| Chronic kidney disease | 5 (13%) | 2 (6%) |
| **Laboratory values** | | |
| C-reactive protein, *mg/L** | 132 [100; 153] | 120 [90; 171] |
| D-dimers, *ng/L* | 831 [630; 1445] | 845 [620; 1900] |
| Ferritin, *µg/L* | 1005 [606; 1456] | 1094 [475; 1606] |
| Lymphocytes count, *G/L* | 0.8 [0.6; 1.1] | 0.9 [0.6; 1.1] |
| Creatinine, *µmol/L* | 80 [64; 104] | 71 [61; 85] |
| **Treatments before randomization** | | |
| Conversion enzyme inhibitor | 8 (22%) | 10 (30%) |
| Antivirals† | 2 (5%) | 0 (0%) |
| Hydroxychloroquine | 0 (0%) | 1 (3%) |
| Azithromycin | 5 (14%) | 4 (12%) |
| Immunosuppressants‡ | 0 (0%) | 1 (3%) |
| Corticosteroids | 20 (54%) | 25 (73%) |
| Corticosteroids ≥ 0.5 mg/kg/d* | 16 (43%) | 20 (59%) |

oSOC = Optimized standard of care group. Values are mean (SD) or median [interquartile range] or number (percentage). SpO = oxygen saturation. BMI = body-mass index. CT = computed tomography.

*Stratification variables for randomization

† lopinavir and ritonavir in both patients

‡ tacrolimus.

## Secondary outcomes

Proportions of treatment success until Day 28 are detailed in Table 3. On Day 3, 10 and 28, the proportion of treatment success was lower in the anakinra group but with no significant difference between the two groups. The evolution of WHO scores over the 28-day follow-up is given in Table 4. We found no significant difference between the two groups except on day 10 where

**Table 2. Number (percentage) of participants alive and without need of mechanical ventilation or ECMO at day 14.**

| D14 success*, n (%) | Anakinra plus oSOC group | oSOC group | Risk difference in percentage points (95%CI) | p |
|---|---|---|---|---|
| Missing data imputed by failure†; $n_1 = 37$, $n_2 = 34$ | 26 (70%) | 31 (91%) | -21 (-39; -2) | 0.027 |
| Complete case analysis; $n_1 = 36$, $n_2 = 34$ | 26 (72%) | 31 (91%) | -19 (-38; -0.1) | 0.041 |
| Adjusted analysis on stratification variables‡ | | | -19 (-38; -0.2) | 0.047 |

*Success was defined as patients being alive and without need of mechanical ventilation or ECMO. $n_1$ corresponds to the number of patients analyzed in the anakinra plus oSOC group, $n_2$ corresponds to the number of patients analyzed in the oSOC group.

†Data was missing for one participant who withdrew consent at day 0 in the anakinra plus oSOC group.

‡ Randomization was stratified on the baseline reactive C-protein level ($<150$ vs. $\geq 150$ mg/L), baseline requirement of oxygen therapy to maintain Sp02 > 92% (3–6 liters per min vs. 7–10 liters per min) and corticosteroid therapy at baseline ($<$ vs. $\geq 0.5$mg/kg/day prednisone).

disease severity classification significantly differed between the two groups (p = 0.008) and we observed a higher proportion of patients hospitalized not requiring supplemental oxygen (WHO score 3) or death (WHO score 7) in the anakinra group than in the oSOC group, respectively 43% vs. 24% and 17% vs. 0%. The evolution of the NEW score over the 28-day follow-up is given in S2 Fig, no difference was found between the two groups; time by treatment interaction 0.02 (95% CI, -0.08 to 0.13, p = 0.67.

After a 28-day follow-up, 9 patients in the anakinra group and 3 in the oSOC group had died. One patient in the oSOC group died after Day 28. Overall survival at day 28 was 75% (95% CI, 62% to 91%) in the anakinra group versus 91% (95% CI, 82% to 100%) (log rank test p = 0.06) (Fig 2A). Causes of death are shown in Table 5. Twenty-one patients were subsequently admitted in ICU: 9 in the anakinra group vs. 13 in the oSOC group (p = 0.16) (Fig 2B). Seven deaths occurred before admission in ICU (6 in the anakinra group vs. 1 in the oSOC group) During the 28-day follow-up, twenty-nine patients needed ventilation support including ECMO, MV, non-invasive ventilation and high flow oxygen: 16 in the anakinra group vs. 13 in oSOC (p = 0.65) (Fig 2C).

The cumulative incidence of patients who withdrew the need for oxygen support at day 28 was 74% (95% CI, 53% to 87%) in the anakinra and 79% (95% CI, 59% to 90%) in the oSOC group, respectively (p = 0.43) (Fig 2D). The cumulative incidence of discharge at day 28 was 69% (95% CI, 51% to 82%) and 68% (95% CI, 49% to 81%] in the anakinra group respectively (p = 0.64) (Fig 2E).

## Biological response

C-reactive protein, ferritin, fibrinogen and D-dimers levels decreased over time in the two groups, while the lymphocyte count increased in both groups (S3 Fig). No between-group difference was found in the evolution of inflammatory parameters.

**Table 3. Number (percentage) of participants alive and without need of mechanical ventilation or ECMO at day 3, 10 and 28.**

| Success*, n (%) | Anakinra plus oSOC group | oSOC group | Risk difference in percentage points (95%CI) | p |
|---|---|---|---|---|
| Day 3; $n_1 = 36$, $n_2 = 34$ | 30 (83%) | 32 (94%) | -11 [-28; 5] | 0.26 |
| Day 10; $n_1 = 36$, $n_2 = 34$ | 27 (75%) | 31 (91%) | -16 [-35; 2] | 0.07 |
| Day 28; $n_1 = 35$, $n_2 = 33$ | 26 (74%) | 29 (88%) | -14 [-33; 6] | 0.15 |

*Success was defined as patients being alive and without need of mechanical ventilation or ECMO. $n_1$ corresponds to the number of patients analyzed in the anakinra plus oSOC group, $n_2$ corresponds to the number of patients analyzed in the oSOC group.

**Table 4. Evolution of WHO scores across 28-day follow-up.**

| WHO score, *n (%)* | Anakinra plus oSOC group | oSOC group | p |
|---|---|---|---|
| **Day 3**; $n_1 = 36$, $n_2 = 34$ | | | 0.36 |
| 1. Not hospitalized, no limitations on activities | 0 (0.0) | 0 (0) | |
| 2. Not hospitalized, limitation on activities | 0 (0.0) | 0 (0) | |
| 3. Hospitalized, not requiring supplemental oxygen | 2 (6) | 0 (0) | |
| 4. Hospitalized, requiring supplemental oxygen | 22 (61) | 25 (74) | |
| 5. Hospitalized, on non-invasive ventilation or high flow oxygen devices | 6 (17) | 7 (21) | |
| 6. Hospitalized, on invasive mechanical ventilation or ECMO | 4 (11) | 2 (6) | |
| 7. Death | 2 (6) | 0 (0) | |
| **Day 10**; $n_1 = 35$, $n_2 = 34$ | | | 0.008 |
| 1. Not hospitalized, no limitations on activities | 1 (3) | 8 (24) | |
| 2. Not hospitalized, limitation on activities | 1 (3) | 3 (9) | |
| 3. Hospitalized, not requiring supplemental oxygen | 15 (43) | 8 (24) | |
| 4. Hospitalized, requiring supplemental oxygen | 8 (23) | 11 (32) | |
| 5. Hospitalized, on non-invasive ventilation or high flow oxygen devices | 1 (3) | 1 (3) | |
| 6. Hospitalized, on invasive mechanical ventilation or ECMO | 3 (9) | 3 (9) | |
| 7. Death | 6 (17) | 0 (0) | |
| **Day 14**; $n_1 = 35$, $n_2 = 32$ | | | 0.29 |
| 1. Not hospitalized, no limitations on activities | 9 (26) | 10 (31) | |
| 2. Not hospitalized, limitation on activities | 12 (34) | 6 (19) | |
| 3. Hospitalized, not requiring supplemental oxygen | 3 (9) | 6 (19) | |
| 4. Hospitalized, requiring supplemental oxygen | 1 (3) | 7 (22) | |
| 5. Hospitalized, on non-invasive ventilation or high flow oxygen devices | 0 (0) | 0 (0) | |
| 6. Hospitalized, on invasive mechanical ventilation or ECMO | 4 (11) | 3 (9) | |
| 7. Death | 6 (17) | 0 (0) | |
| **Day 28**; $n_1 = 35$, $n_2 = 32$ | | | 0.27 |
| 1. Not hospitalized, no limitations on activities | 12 (34) | 15 (47) | |
| 2. Not hospitalized, limitation on activities | 12 (34) | 6 (19) | |
| 3. Hospitalized, not requiring supplemental oxygen | 2 (6) | 5 (16) | |
| 4. Hospitalized, requiring supplemental oxygen | 0 (0) | 2 (6) | |
| 5. Hospitalized, on non-invasive ventilation or high flow oxygen devices | 0 (0) | 0 (0) | |
| 6. Hospitalized, on invasive mechanical ventilation or ECMO | 0 (0) | 1 (3) | |
| 7. Death | 9 (26) | 3 (9) | |

ECMO = Extracorporeal membrane oxygenation.

Values are number (percentage).

## Safety

A total of 32 (87%) out of 37 patients in the anakinra group and 22 (65%) in the oSOC group reported at least one adverse event (p = 0.016) including 19 (51%) patients in the anakinra group and 18 (53%) in the oSOC group (p = 0·89) who presented at least one serious adverse event (Table 5). Altogether, 59 adverse events occurred in the anakinra group with 9 events (15%) related to anakinra (adverse effects) and 50 adverse events in the oSOC group.

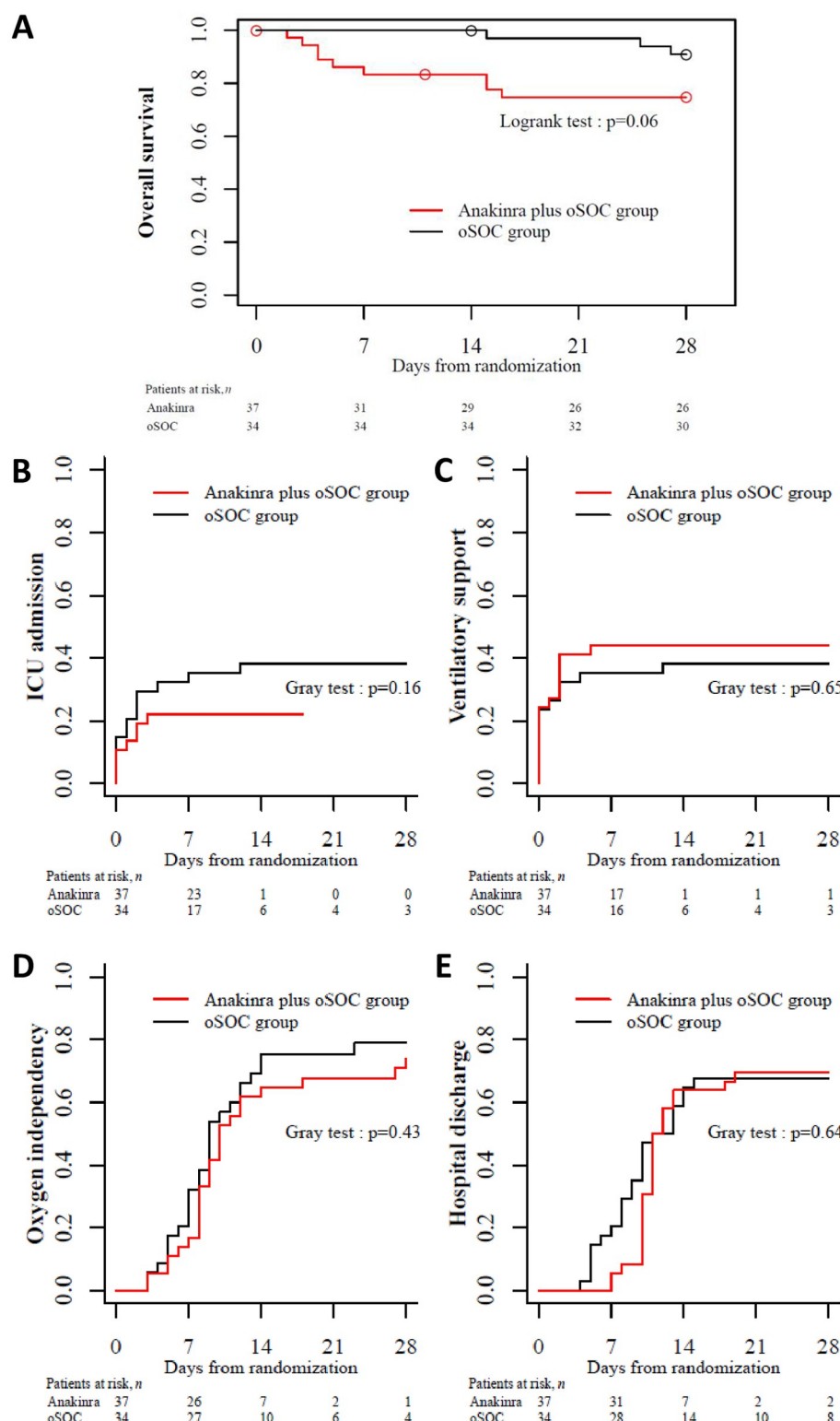

**Fig 2. Secondary outcomes for analyzed patients over 28 days' follow-up. A**, Overall survival; **B**, Cumulative incidence of ICU admission; **C**, Cumulative incidence of need for ventilatory support; **D**, Cumulative incidence of oxygen independency; and **E**, Cumulative incidence of hospital discharge. ICU, intensive care unit; oSOC, optimized standard of care.

**Table 5. Adverse events, serious adverse events and causes of death.**

| | Anakinra plus oSOC group (n = 37) | oSOC group (n = 34) | p |
|---|---|---|---|
| **Adverse events** | | | |
| Patients with at least one adverse event | 32 (87%) | 22 (65%) | 0.03 |
| Patients with multiple adverse events | 13 (35%) | 14 (41%) | 0.60 |
| Total number of adverse events | 59 | 50 | |
| Adverse event related to study treatment | 9 (24%) | 0 | |
| **Serious adverse events** | | | |
| Patients with at least one serious adverse event | 19 (51%) | 18 (53%) | 0.89 |
| Patients with multiple serious adverse events | 1 | 1 | |
| Total number of serious adverse events | 22 | 19 | |
| ARDS | 14 | 13 | |
| Pulmonary embolism | 1 | 2 | |
| Bacterial pneumoniae | 1 | 0 | |
| Hepatic cytolysis | 1 | 2 | |
| Digestive tract haemorrhage | 0 | 1 | |
| Anakinra erratum dosage (without adverse effect) | 2 | 0 | |
| Metastatic progression | 0 | 1 | |
| Renal failure | 1 | 0 | |
| Vasoplegia syndrome | 1 | 0 | |
| Sudden death | 1 | 0 | |
| **Cause of deaths** | 9 (24%) | 4 (12%) | 0.27 |
| ARDS | 7 | 3 | |
| Metastatic progression | 0 | 1 | |
| Sudden death | 1 | 0 | |
| Mesenteric ischemia | 1 | 0 | |

ARDS = Acute respiratory distress syndrome.

Values are number (percentage).

The nine events related to anakinra occurred in 6 patients, 3 patients presented serious adverse effects: 1 hepatic cytolysis, 1 acute respiratory distress syndrome (dubious relationship) and 1 sudden death (dubious relationship), and 3 patients presented non-serious events as assessed by the local investigators: 1 hepatic cytolysis, 1 renal failure, 1 abdominal pain, 1 hepatomegaly, 1 cystitis and 1 paresthesia.

## Discussion

In this trial, we found success proportion to be significantly lower in the anakinra than in the oSOC group for patients with moderate COVID-19 pneumonia. The trial ended prematurely due to higher mortality rates (although not statistically significant) in the anakinra group than in the oSOC group. Secondary outcomes are also in favor of a better outcome in the oSOC group than in the anakinra group. To our knowledge our study is the first to suggest a negative effect of anakinra in COVID-19.

At the beginning of the COVID-19 pandemic, months before the RECOVERY trial demonstrated DXM efficacy, no standard of care had rapidly emerged and anakinra was thought to be a promising drug. IL1-1β was promptly described as one of the pivotal cytokines involved in the viral driven cytokine storm. Indeed, IL1-1β is strongly increased in both the serum and

bronchial aspiration of COVID 19 patients [5]. Thus, anakinra as a recombinant, anti-human IL-1 receptor treatment and well-known drug seemed an attractive target to test thus explaining the rapid construction of the ANACONDA trial. Rapidly following the initiation of ANACONDA, some observational studies, i.e., a French and an Italian, monocentric case series, suggested the possible efficacy of anakinra for patients with mild-to-moderate, severe, or critical COVID-19 [12, 18]. However, the main limitation of these case series was the comparison of patients treated off label with anakinra to "historical patients", managed at the very beginning of the first wave of the pandemic. At that time, mortality was high, at around 50% which could be explained by no available standard of care [12]. Thus, anakinra efficacy could have been overestimated and it was crucial to conduct robust randomized controlled trials to obtain an unbiased treatment effect of anakinra.

So far, the only other published RCT evaluating anakinra efficacy in COVID-19 patients was stopped for futility. This trial, led by the CORIMUNO group, showed that anakinra did not improve outcomes in patients with mild-to-moderate COVID-19 pneumonia [15]. The two co-primary outcomes were the proportion of patients who had died or needed non-invasive or mechanical ventilation by day 4 and survival without the need for mechanical or non-invasive ventilation at day 14. The patients enrolled were comparable to those enrolled in our trial regarding baseline demographic characteristics (age, sex, body mass index, comorbidities) and baseline COVID-19 pneumonia severity (C-reactive protein, oxygen flow, respiratory rate). More recently, the SAVE MORE study suggest a potential beneficial effect of anakinra in moderate COVID-19 patients [18]. The SAVE MORE study is a double-blind, randomized controlled trial evaluated the efficacy and safety of anakinra as compared to SOC in 594 patients with COVID-19 at risk of progressing to respiratory failure. To be included patients had to have plasma soluble urokinase plasminogen activator receptor (suPAR) $\geq$6 ng ml$^{-1}$. Early increase suPAR serum levels is indicative of increased risk of progression of COVID-19 to respiratory SDRA. This study found a decreased 28-day mortality and shorter hospital stay in the anakinra group. These results are in discrepancy with the two main other RCTs and may be explained by the difference in selection criteria and a more severe population of patients in the SAVE-MORE trial. Indeed, anakinra could be efficient in a subgroup of moderate COVID-19. In France, suPAR is not available in daily practice and has not been monitored in the patients enrolled in ANACONDA. Another explanation of this discrepancy could be the different dosage used (higher in our study) and type of anakinra application (IV in our study versus SC in the SAVE MORE study)."

Contrary to our hypothesis, and in disagreement with the CORIMUNO RCT, we observed a significantly lower proportion of patients on ECMO, mechanical ventilation or who died by day 14 in the oSOC group than in the anakinra group and overall survival until day 28 was higher in the oSOC group than in the anakinra group. In the ANACONDA RCT, the vast majority of deaths, in both groups, were linked to the natural evolution of the disease (ARDS). No death was directly linked to anakinra side effects or bacterial infections, all patients were treated with antibiotics (inclusion criteria). Despite stratification of randomization on corticosteroid therapy at baseline ($<$ vs. $\geq$ 0.5mg/kg/day prednisone), a greater proportion of patients randomized in the oSOC group received corticosteroids as compared to those in the anakinra group. Nevertheless, analysis of the primary outcome adjusted on stratification variables (including corticosteroid therapy at baseline) showed similar results to unadjusted ones. Our results could also be explained by a deleterious effect of anakinra i.e., by suppressing the immunological response during high replication of the virus (viral load was not monitored). The main difference between the CORIMUNO RCT and our trial was the duration of anakinra administration, respectively 5 vs. 10 days. At this time, at least 10 RCTs testing anakinra in COVID-19 patients are ongoing and should soon provide more evidence of the effect of anakinra.

Our study has several strengths, including its robust randomized controlled trial and multi-centre design. Limitations of the trial include the absence of blinding. The trial was not blinded because it was logistically impossible at the time of the pandemic to produce a placebo and to set up a double-blind study in due time. Another limitation is the trial sample. Indeed, 71 patients were enrolled instead of the 240 planned. However, as previously mentioned the trial stopped prematurely for safety concerns.

In summary, contrary to our hypothesis, we found that anakinra was inferior to oSOC in patients with moderate COVID-19 pneumonia. Thus, we do not recommend the use of anakinra in all patients with moderate COVID-19 pneumonia without having monitored the suPAR level.

## Supporting information

**S1 File. Trial protocol.**
(DOC)

**S2 File. Final statistical analysis plan.**
(DOCX)

**S3 File. Consort 2010 statement.**
(PDF)

**S1 Table. Anakinra administration, number of doses administered per patient and reasons for incomplete regimen.**
(DOCX)

**S2 Table. Corticoids administered at inclusion or during follow-up.**
(DOCX)

**S1 Fig. Subgroup analyses of the primary outcome.**
(DOCX)

**S2 Fig. Evolution of NEWs score across the 28-day follow-up.**
(DOCX)

**S3 Fig. Evolution of inflammatory parameters across the 28-day follow-up.**
(DOCX)

## Acknowledgments

We warmly thank Elodie Mousset for her precious help concerning the coordination of the trial. We thank the patients and their family for their willingness to participate in the trial. We thank all the members of the data safety monitoring board (DSMB): Dr. Bénédicte Lebrun-Vignes, Prof. Sophie Georgin-Lavialle, Dr. Jean-Benoit Hardouin and Prof. Renaud Verdon. We also kindly thank Frédérique Musset and Maeva Dieu for their helpful advice and work. We greatly thank Daniel Audemard, Patrick Coupier and Michèle Cohen for their help in collecting funds to support the trial.

## Author Contributions

**Conceptualization:** Alexandra Audemard-Verger, Amélie Le Gouge, Marc-Olivier Vareil, Radjiv Goulabchand, Léa Colombain, Adrien Bigot, Thomas Guimard, Youcef Douadi, Geoffrey Urbanski, Jean François Faucher, Laurence Maulin, Bertrand Lioger, Joseph

Emmerich, Ann-Rose Cook, Hélène Bourgoin, Arsène Mekinian, François Maillot, Agnès Caille.

**Data curation:** Alexandra Audemard-Verger, Amélie Le Gouge, Marc-Olivier Vareil, Mathilde Devaux, Boris Bienvenu, Vincent Leroy, Radjiv Goulabchand, Léa Colombain, Adrien Bigot, Thomas Guimard, Jean François Faucher, Bertrand Lioger, Matthieu Groh, Joseph Emmerich, Sophie Deriaz, Arsène Mekinian, Achille Aouba, François Maillot, Agnès Caille.

**Formal analysis:** Alexandra Audemard-Verger, Amélie Le Gouge, Vincent Pestre, Johan Courjon, Marc-Olivier Vareil, Mathilde Devaux, Boris Bienvenu, Adrien Bigot, Thomas Guimard, Youcef Douadi, Geoffrey Urbanski, Jean François Faucher, Bertrand Lioger, Jean-Philippe Talarmin, Matthieu Groh, Joseph Emmerich, Sophie Deriaz, Nicole Ferreira-Maldent, Ann-Rose Cook, Arsène Mekinian, Achille Aouba, François Maillot.

**Funding acquisition:** Alexandra Audemard-Verger, Johan Courjon, Vincent Langlois, Vincent Leroy, Radjiv Goulabchand, Léa Colombain, Adrien Bigot, Thomas Guimard, Geoffrey Urbanski, Jean François Faucher, Laurence Maulin, Bertrand Lioger, Jean-Philippe Talarmin, Matthieu Groh, Joseph Emmerich, Nicole Ferreira-Maldent, Hélène Bourgoin, Arsène Mekinian.

**Investigation:** Alexandra Audemard-Verger, Marc-Olivier Vareil, Radjiv Goulabchand, Léa Colombain, Adrien Bigot, Jean François Faucher, Jean-Philippe Talarmin, Matthieu Groh, Joseph Emmerich, Ann-Rose Cook, Céline Lengellé, François Maillot.

**Methodology:** Alexandra Audemard-Verger, Youcef Douadi, Nicole Ferreira-Maldent, Céline Lengellé, Achille Aouba, Agnès Caille.

**Project administration:** Alexandra Audemard-Verger, Vincent Leroy, Youcef Douadi, Nicole Ferreira-Maldent, Ann-Rose Cook, Achille Aouba, Agnès Caille.

**Resources:** Alexandra Audemard-Verger, Vincent Leroy, Céline Lengellé.

**Software:** Johan Courjon, Hélène Bourgoin.

**Supervision:** Alexandra Audemard-Verger, Amélie Le Gouge, Johan Courjon, Nicole Ferreira-Maldent, Hélène Bourgoin.

**Validation:** Alexandra Audemard-Verger, Amélie Le Gouge, Vincent Pestre, Vincent Langlois, Boris Bienvenu, Geoffrey Urbanski, Nicole Ferreira-Maldent, Céline Lengellé, Hélène Bourgoin, François Maillot.

**Visualization:** Alexandra Audemard-Verger, Amélie Le Gouge, Nicole Ferreira-Maldent, Céline Lengellé.

**Writing – original draft:** Alexandra Audemard-Verger, Amélie Le Gouge, Vincent Pestre, Vincent Langlois, Boris Bienvenu, Geoffrey Urbanski, Laurence Maulin, Sophie Deriaz, Nicole Ferreira-Maldent, Ann-Rose Cook, Agnès Caille.

**Writing – review & editing:** Alexandra Audemard-Verger, Amélie Le Gouge, Vincent Pestre, Vincent Langlois, Boris Bienvenu, Laurence Maulin, Sophie Deriaz, Arsène Mekinian, Agnès Caille.

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
