## [Decision Letter · Decision Letter 0]

16 Feb 2022

PONE-D-21-35547Efficacy and Safety of Anakinra in Adults presenting deteriorating Respiratory Symptoms fromCOVID-19: A randomized controlled trialPLOS ONE

Dear Dr. Audemard-Verger,

Thank you for submitting your manuscript to PLOS ONE. After careful consideration, we feel that it has merit but does not fully meet PLOS ONE’s publication criteria as it currently stands. Therefore, we invite you to submit a revised version of the manuscript that addresses the points raised during the review process.

Dear authors, please revise the manuscript according to reviewers suggestions or write a detailed rebuttal on a point by point basis. 

We look forward to receiving your revised manuscript.

Kind regards,

Davor Plavec, MD, MSc, PhD, Prof.

Academic Editor

PLOS ONE

Journal Requirements:

2. During your revisions, please note that a simple title correction is required: Please insert a space before COVID-19 so that your title reads "Efficacy and Safety of Anakinra in Adults presenting deteriorating Respiratory Symptoms from COVID-19: A randomized controlled trial". Please ensure this is updated in the manuscript file and the online submission information.

SOBI

Fonds de doatation CHRU de Tours

Sobi and Fond de Dotation CHRU de Tours

SOBI

Fonds de doatation CHRU de Tours

6. Thank you for stating the following in your Competing Interests section:  

None

7. Please note that in order to use the direct billing option the corresponding author must be affiliated with the chosen institute. Please either amend your manuscript to change the affiliation or corresponding author, or email us at plosone@plos.org with a request to remove this option.

8. We note that you have stated that you will provide repository information for your data at acceptance. Should your manuscript be accepted for publication, we will hold it until you provide the relevant accession numbers or DOIs necessary to access your data. If you wish to make changes to your Data Availability statement, please describe these changes in your cover letter and we will update your Data Availability statement to reflect the information you provide.

9. Please include a caption for figure 2.

10. Please include captions for your Supporting Information files at the end of your manuscript, and update any in-text citations to match accordingly. Please see our Supporting Information guidelines for more information: http://journals.plos.org/plosone/s/supporting-information. 

Additional Editor Comments:

Dear authors, please revise the manuscript according to reviewers suggestions or write a detailed rebuttal on a point by point basis.

Reviewers' comments:

Reviewer's Responses to Questions

**Comments to the Author**

1. Is the manuscript technically sound, and do the data support the conclusions?

Reviewer #1: Yes

Reviewer #2: Yes

Reviewer #3: Partly

2. Has the statistical analysis been performed appropriately and rigorously? 

Reviewer #1: Yes

Reviewer #2: No

Reviewer #3: I Don't Know

3. Have the authors made all data underlying the findings in their manuscript fully available?

Reviewer #1: Yes

Reviewer #2: No

Reviewer #3: Yes

4. Is the manuscript presented in an intelligible fashion and written in standard English?

Reviewer #1: Yes

Reviewer #2: Yes

Reviewer #3: Yes

5. Review Comments to the Author

Reviewer #1: I beleive that this article should be fast tracked for publishing. I can only suggest rechecking of the spaces in the text, in my copy some spaces are missing so the proper lecture of the article would be of use but the medical information is important and urgent.

Reviewer #2: This robust study is extremely well reported and mostly well analysed.

My main concern:

Linear models are not appropriate for binary outcomes, such as the primary endpoint in this study. The authors should refit the primary outcome models adjusting for stratification variables using logistic regression, and the same for their subgroup analyses.

Other minor questions/corrections:

Were the analysts or the people collecting the follow-up data blinded to the treatment allocation?

On page 8, sites clinicians should be written as sites’ clinicians.

How was the dose decided upon for these patients?

The units for the risk difference should be stated in Figure S2.

Some interpretation/explanation for the parameters in the tables accompanying Figures S3 and S4 should be given.

The term ‘Dubious relationship’ between adverse events and the treatment is ambiguous and not standard. It would be better if the authors used the standard terminology ie. ‘Unlikely related’, ‘Possible relationship’, ‘Probable relationship’ etc.

First paragraph of the discussion - ‘negatif’ should say ‘negative’ and in the second paragraph ‘aFrench’ should say ‘a French’.

The figure legend at the top of page 16 is actually for Figure 2, not Figure 1 as stated, and there is no legend for Figure 1.

Reviewer #3: 1. Objective of this study was to investigate whether anakinra improves outcome in COVID-19 patients (assess the efficacy of Anakinra+optimized Standard of Care (oSOC) as compared to oSOC alone on condition of patients with COVID -19 infection and worsening respiratory symptoms).

In conclusion, Anakinra was inferior to oSOC in patients with moderate COVID-19 pneumonia.

The conclusion was not in line with the suggested objective. The authors did not specify moderate COVID in the study objective.

2. In discussion, the authors mentioned that the only other published RCT evaluating anakinra efficacy in COVID-19 patients was stopped for futility (CORIMUNO trial). They did not comment SAVE-MORE study (which is a pivotal, confirmatory, prospective, multicenter, double-blind, randomized, placebo-controlled study in hospitalized patients with confirmed infection with SARS-CoV-2, LRTI, and plasma suPAR levels ≥6 ng/ml). In the contrast with the SAVE-MORE study, the ANACONDA study is terminated earlier with safety concerns. They mentioned different biological markers (CRP, ferritin, fibrinogen, D-dimers, IL-6) but not suPAR. A suPAR is a biomarker of early deterioration of patients which integrates information on 3 different functions (i.e. inflammatory cascade activation, coagulation, and endothelial-neutrophil interaction). These three functions are up-regulated in COVID-19, which explains why using biomarkers that only carry information for one of the functions cannot provide the integrated information as suPAR does. It seems, according to the SAVE-MORE study as well as the last changes in Anakinra regulatory approval, that suPAR is a crucial biomarker concerning to assessment of COVID progression to respiratory failure.

3. Authors did not comment on the possible influence of dose (higher than used in SAVE-MORE study) type of anakinra application (intravenous vs subcutaneous) on the study results.

4. Results of this study are completely in opposition to the decision of EMAs CHMP from December last year without giving possible explanation or reasons (e.g. Anakinra has effect only in patients with suPAR level ≥6 ng/ml).

5. The typographical errors should be corrected (e.g. futhemore, negatif, space between the words-aFrench).

6. PLOS authors have the option to publish the peer review history of their article (what does this mean?). If published, this will include your full peer review and any attached files.

Reviewer #1: No

Reviewer #2: **Yes: **Sarah J.E. Barry

Reviewer #3: No

---

## [Author Response · Author response to Decision Letter 0]

20 Apr 2022

Tours, March 2022

Dear Editor,

 You will find enclosed a revised version of our manuscript (PONE-D-21-35547) entitled: “Efficacy and Safety of Anakinra in Adults presenting deteriorating Respiratory Symptoms from COVID-19: A randomized controlled trial" that we would like to resubmit to Plos One.

 We thank the Reviewers for their helpful comments. You will find below the responses to the Reviewers’ comments, a marked version of the manuscript made from the previous version and a clean version. We have addressed their comments and made the appropriate modifications.

 We hope that our manuscript will be now suitable for publication in Plos One.

Sincerely yours,

Dr Alexandra Audemard-Verger and Dr Agnès Caille

Author’s response: We have performed some changes to meet PLOS ONE's style requirements (figure citation, level heading, and reference citations…)

2. During your revisions, please note that a simple title correction is required: Please insert a space before COVID-19 so that your title reads "Efficacy and Safety of Anakinra in Adults presenting deteriorating Respiratory Symptoms from COVID-19: A randomized controlled trial". Please ensure this is updated in the manuscript file and the online submission information.

Author’s response: We agree and added a space before COVID 19.

SOBI

Fonds de dotation CHRU de Tours

Author’s response: We added a paragraph in the financial disclosure “The funders had no role in study design, data collection and analysis, decision to publish, or preparation of the manuscript." 

Sobi and Fond de Dotation CHRU de Tours

SOBI

Fonds de dotation CHRU de Tours

Author’s response: We removed of the Acknowledgments section Sobi and Fond de Dotation CHRU de Tours.

Author’s response: We will correct it when resubmitting.

6. Thank you for stating the following in your Competing Interests section: 

None

Author’s response: We will do it.

7. Please note that in order to use the direct billing option the corresponding author must be affiliated with the chosen institute. Please either amend your manuscript to change the affiliation or corresponding author, or email us at plosone@plos.org with a request to remove this option.

8. We note that you have stated that you will provide repository information for your data at acceptance. Should your manuscript be accepted for publication, we will hold it until you provide the relevant accession numbers or DOIs necessary to access your data. If you wish to make changes to your Data Availability statement, please describe these changes in your cover letter and we will update your Data Availability statement to reflect the information you provide.

Author’s response: We do not want to modify it, data will be available by contacting the first author. We will not deposit our data on a repository space.

9. Please include a caption for figure 2. 

Author’s response: We added a caption for figure 2.

10. Please include captions for your Supporting Information files at the end of your manuscript, and update any in-text citations to match accordingly. Please see our Supporting Information guidelines for more information: http://journals.plos.org/plosone/s/supporting-information. 

Author’s response: We added a caption for the 3 Supporting Information Files at the end of the manuscript and have updated in-text citations.

Reviewers' comments:

Reviewer #1: I beleive that this article should be fast tracked for publishing. I can only suggest rechecking of the spaces in the text, in my copy some spaces are missing so the proper lecture of the article would be of use but the medical information is important and urgent.

Author’s response: We warmly thank reviewer 1 for this kind comment.

Reviewer #2: This robust study is extremely well reported and mostly well analysed.

My main concern:

Linear models are not appropriate for binary outcomes, such as the primary endpoint in this study. The authors should refit the primary outcome models adjusting for stratification variables using logistic regression, and the same for their subgroup analyses.

Author’s response: We appreciate this suggestion. Nevertheless, because our sample size was based on an hypothesized risk difference, it seems to us more relevant to present our results in the same way. For adjusted analyses on stratification variables, we used linear regression models (identity response function and normal distribution) to obtain an adjusted risk difference through the estimated treatment effect. Indeed, this approach is one of the possible and performant approach to estimate adjusted risk differences [ref] (Pedroza, C., Truong, V.T. Performance of models for estimating absolute risk difference in multicenter trials with binary outcome. BMC Med Res Methodol 16, 113 (2016).). However, to address the reviewer’s concern and in accordance with the CONSORT statement, the relative treatment effect for the primary outcome (odds ratio) was added in the manuscript. 

Results for the primary outcome is now as follows: “On day 14, the proportion of treatment success was significantly lower in the anakinra group 70% (n=26) vs. 91% (n=31) in the oSOC group; risk difference -21 percentage points; (95% CI, -39 to -2), odds ratio 0.23 (95% CI, 0.06 to 0.91), p=0.027 (Table 2).”

Other minor questions/corrections:

Were the analysts or the people collecting the follow-up data blinded to the treatment allocation? 

Author’s response: Outcome assessors and analysts were not blinded to the treatment allocation. We added this information in the Methods section of the revised manuscript under Randomization and blinding “Study physicians, research staff, participants, and data analysts were aware of treatment allocation.”

On page 8, sites clinicians should be written as sites’ clinicians.

Author’s response: We agree and have corrected this mistake.

How was the dose decided upon for these patients? 

Author’s response: Anakinra is currently approved for chronic treatment in several inflammatory diseases mostly at dose of 100 mg/day or 200 mg/day. So far, anakinra efficacy and dose to reach efficacy was unknown in COVID-19 patients. In order to select the appropriate dose of anakinra, we set up a Scientific Committee composed of French experts who prescribe anakinra in inflammatory diseases: Prof. Sophie GEORGIN-LAVIALLE, Internal Medicine Department, Hospital Tenon, Paris and Prof. Arsène MEKINIAN, Internal Medicine Department, Hospital Saint Antoine, Paris. Prof. Sophie GEORGIN-LAVIALLE coordinates the French Center of auto-inflammatory diseases (Centre de Références des Maladies AutoInflammatoires et des Amyloses inflammatoires (CEREMAIA)) and has collaborated to the development of the Nation Health Protocol (PNDS) of Still Adult Disease. The Scientific Committee also solicited French physicians who had treated COVID-19 patients with anakinra (off label) successfully including Prof. Joseph EMMERICH, Hospital Saint Joseph, Paris. Thus, Anakinra dose in this trial was decided based on the Scientific Committee advice and considering benefits/risks balance.

The units for the risk difference should be stated in Figure S2. 

Author’s response: We now state that “Risk difference was defined as the between-group difference (Anakinra group minus Standard care group) in percentage points.” in Figure S2

Some interpretation/explanation for the parameters in the tables accompanying Figures S3 and S4 should be given. 

Author’s response: We modified parameters labels in tables accompanying Figures S3 and S4 to clarify interpretation and added explanations in figure S3:”Between group difference in NEW score evolution was assessed through time by treatment interaction term” and in figure S4:”Between-group differences in inflammatory parameters evolutions were assessed through time by treatment interaction term.”. We also added more details in the section results of the manuscript:” time by treatment interaction 0.02 (95% CI, -0.08 to 0.13)”.

The term ‘Dubious relationship’ between adverse events and the treatment is ambiguous and not standard. It would be better if the authors used the standard terminology ie. ‘Unlikely related’, ‘Possible relationship’, ‘Probable relationship’ etc. 

Author’s response: We disagree with the reviewer’s comment. The term ‘Dubious relationship’ is recommended by the French methodology of pharmacovigilance when the relationship is unclear. The reference of this method is “ Ghada Miremont-Salamé, Hélène Théophile, Françoise Haramburu, Bernard Bégaud, Imputabilité en pharmacovigilance : de la méthode française originelle aux methods réactualisées,Therapies,Volume 71, Issue 2,2016,doi.org/10.1016/j.therap.2016.02.009”

First paragraph of the discussion - ‘negatif’ should say ‘negative’ and in the second paragraph ‘aFrench’ should say ‘a French’. 

Author’s response: We agree and have corrected these mistakes.

The figure legend at the top of page 16 is actually for Figure 2, not Figure 1 as stated, and there is no legend for Figure 1. 

Author’s response: We agree and have corrected this mistake.

Reviewer #3: 1. Objective of this study was to investigate whether anakinra improves outcome in COVID-19 patients (assess the efficacy of Anakinra+optimized Standard of Care (oSOC) as compared to oSOC alone on condition of patients with COVID -19 infection and worsening respiratory symptoms).

In conclusion, Anakinra was inferior to oSOC in patients with moderate COVID-19 pneumonia.

The conclusion was not in line with the suggested objective. The authors did not specify moderate COVID in the study objective. 

Author’s response: We agree and have added the term moderate in the objective

2.3. 4 In discussion, the authors mentioned that the only other published RCT evaluating anakinra efficacy in COVID-19 patients was stopped for futility (CORIMUNO trial). They did not comment SAVE-MORE study (which is a pivotal, confirmatory, prospective, multicenter, double-blind, randomized, placebo-controlled study in hospitalized patients with confirmed infection with SARS-CoV-2, LRTI, and plasma suPAR levels ≥6 ng/ml). In the contrast with the SAVE-MORE study, the ANACONDA study is terminated earlier with safety concerns. They mentioned different biological markers (CRP, ferritin, fibrinogen, D-dimers, IL-6) but not suPAR. A suPAR is a biomarker of early deterioration of patients which integrates information on 3 different functions (i.e. inflammatory cascade activation, coagulation, and endothelial-neutrophil interaction). These three functions are up-regulated in COVID-19, which explains why using biomarkers that only carry information for one of the functions cannot provide the integrated information as suPAR does. It seems, according to the SAVE-MORE study as well as the last changes in Anakinra regulatory approval, that suPAR is a crucial biomarker concerning to assessment of COVID progression to respiratory failure. Authors did not comment on the possible influence of dose (higher than used in SAVE-MORE study) type of anakinra application (intravenous vs subcutaneous) on the study results. 

4. Results of this study are completely in opposition to the decision of EMAs CHMP from December last year without giving possible explanation or reasons (e.g. Anakinra has effect only in patients with suPAR level ≥6 ng/ml). 

Author’s response: We agree and added a paragraph in the discussion section “More recently, the SAVE MORE study suggest a potential beneficial effect of anakinra in moderate COVID-19 patients �18�.. The SAVE MORE study is a double-blind, randomized controlled trial evaluated the efficacy and safety of anakinra as compared to SOC in 594 patients with COVID-19 at risk of progressing to respiratory failure. To be included patients had to have plasma soluble urokinase plasminogen activator receptor (suPAR) ≥6 ng ml-1. Early increase suPAR serum levels is indicative of increased risk of progression of COVID-19 to respiratory SDRA. This study found a decreased 28-day mortality and shorter hospital stay in the anakinra group. These results are in discrepancy with the two main other RCTs and may be explained by the difference in selection criteria and a more severe population of patients in the SAVE-MORE trial. Indeed, anakinra could be efficient in a subgroup of moderate COVID-19. In France, suPAR is not available in daily practice and has not been monitored in the patients enrolled in ANACONDA. Another explanation of this discrepancy could be the different dosage used (higher in our study) and type of anakinra application (IV in our study versus SC in the SAVE MORE study).”

5. The typographical errors should be corrected (e.g. futhemore, negatif, space between the words-aFrench).

 � Author’s response: We agree and have corrected the typos

---

## [Decision Letter · Decision Letter 1]

16 May 2022

Efficacy and Safety of Anakinra in Adults presenting deteriorating Respiratory Symptoms from COVID-19: A randomized controlled trial

PONE-D-21-35547R1

Dear Dr. Audemard-Verger,

We’re pleased to inform you that your manuscript has been judged scientifically suitable for publication and will be formally accepted for publication once it meets all outstanding technical requirements.

Kind regards,

Davor Plavec, MD, MSc, PhD, Prof.

Academic Editor

PLOS ONE

Additional Editor Comments (optional):

As all concerns of the reviewers are addressed the manuscript is acceptable in its current form.

Reviewers' comments:

Reviewer's Responses to Questions

**Comments to the Author**

1. If the authors have adequately addressed your comments raised in a previous round of review and you feel that this manuscript is now acceptable for publication, you may indicate that here to bypass the “Comments to the Author” section, enter your conflict of interest statement in the “Confidential to Editor” section, and submit your "Accept" recommendation.

Reviewer #1: All comments have been addressed

Reviewer #2: All comments have been addressed

Reviewer #3: All comments have been addressed

2. Is the manuscript technically sound, and do the data support the conclusions?

Reviewer #1: Yes

Reviewer #2: Yes

Reviewer #3: Yes

3. Has the statistical analysis been performed appropriately and rigorously? 

Reviewer #1: I Don't Know

Reviewer #2: Yes

Reviewer #3: Yes

4. Have the authors made all data underlying the findings in their manuscript fully available?

Reviewer #1: Yes

Reviewer #2: No

Reviewer #3: Yes

5. Is the manuscript presented in an intelligible fashion and written in standard English?

Reviewer #1: Yes

Reviewer #2: Yes

Reviewer #3: Yes

6. Review Comments to the Author

Reviewer #1: As the first time I have no specific recommendations for further changes. I believe that this article should be fast tracked for publishing.

Reviewer #2: (No Response)

Reviewer #3: Thank you for your response. In my opinion, all major and minor comments have been adequately addressed.

7. PLOS authors have the option to publish the peer review history of their article (what does this mean?). If published, this will include your full peer review and any attached files.

Reviewer #1: No

Reviewer #2: No

Reviewer #3: No

---

## [Editor Report · Acceptance letter]

27 Jul 2022

PONE-D-21-35547R1 

Efficacy and Safety of Anakinra in Adults presenting deteriorating Respiratory Symptoms from COVID-19:  A randomized controlled trial 

Dear Dr. Audemard-Verger:

I'm pleased to inform you that your manuscript has been deemed suitable for publication in PLOS ONE. Congratulations! Your manuscript is now with our production department. 

Kind regards, 

on behalf of

Dr. Davor Plavec 

Academic Editor

PLOS ONE